# Comparison of self-reported survey and wastewater-based epidemiology measures of cocaine use on a college campus

Shona McCulloch[1], Dessa Bergen-Cico[1], Teng Zeng[2], David A. Larsen[1]*

1 Department of Public Health, Syracuse University, Syracuse, New York, United States of America,
2 Department of Civil and Environmental Engineering, Syracuse University, Syracuse, New York, United States of America

* dalarsen@syr.edu

## Abstract

### Background

Wastewater–based epidemiology (WBE) has the potential to produce reliable, efficient, and non-invasive measures of current psychoactive drug use. The aim of this study was to assess the feasibility and validity of using WBE to estimate current cocaine use among university students at a residential campus.

### Methods

We analyzed wastewater samples from four locations at a residential university campus during spring semester of 2021, testing for cocaine metabolites in addition to control comparison substances (acetaminophen and caffeine). We simultaneously administered a confidential self-report survey of recent substance use behaviors to a randomized sample of undergraduate students at this university.

### Results

Self-reported survey estimates of cocaine use and point estimates of cocaine use derived from wastewater-based epidemiology are similar, but the survey is imprecise with a wide CI, and agreement is sensitive to key WBE assumptions; thus, results are consistent but not conclusive. The self-report survey results indicated 0.13% of respondents were regular cocaine users, which is equivalent to the estimate of 0.12% of students using cocaine as measured through WBE. This prevalence is also in line with the 0.14% National American College Health Association (NACHA) survey during the same semester.

**Data availability statement:** All relevant data are within the manuscript and its Supporting information files.

**Funding:** TZ received funding from the National Science Foundation, award number 2018497.

**Competing interests:** The authors have declared that no competing interests exist.

## Conclusions

WBE shows promise as a complementary approach for estimating current cocaine use among students on a residential campus; with current data the WBE point estimate is similar to the survey point estimate, but uncertainty in both measures (especially the survey) requires further research.

## Introduction

Cocaine is an illicit stimulant drug with a high addiction potential due to its effects on the brain's dopaminergic pathways [1]. In addition to its risk for addiction cocaine, which is most commonly taken in powder form intranasally, cocaine can be mixed with other drugs such as amphetamines and synthetic opioids, including fentanyl, increasing the danger to users. These "polydrug cocaine combinations" that emerged concurrently with the fentanyl epidemic in 2014 in the United States are largely responsible for cocaine related overdose deaths [2]. According to the National Institute of Health (NIH) cocaine attributable deaths in the United States (U.S.) rose 54% from 2019 through 2021 with 24,486 total deaths in 2021 [3].

Historically, North America has been the largest market in the world for cocaine with an estimated 6.4 million users in 2020; however global cocaine trafficking and use has increased in recent years negatively impacting global health and playing a significant role in incarcerations and organized crime [4–6]. Approximately 2% of the U.S. population 12 years and older have used cocaine [5], and an estimated 1.3 million Americans struggle with cocaine use disorder (CUD) [7]. Young adults aged 18–25 years old have the highest level of cocaine use [8]. Consequently, there is a growing body of research encompassing both illicit drug (ID) and prescription drug misuse (PDM) on college campuses. University life presents multiple stressors, including academic pressure, social challenges, and financial concerns, which may contribute to substance use among students [9].

Cocaine use on college campuses varies by institution and state and is influenced by a variety of factors including physical and social environment, availability, and purpose for use (i.e., stress relief, recreation, compulsion etc.) [9]. Although the long-term impacts of repeated cocaine use are not yet fully realized [10], chronic cocaine use has been shown to cause serious cognitive deficits in working memory [11], impulsivity and control [12], and disrupted midbrain functionality [13]. Brain development during formative college years is critical and could potentially be adversely impacted by cocaine use. Therefore, understanding the extent of cocaine use on college campuses can inform public health interventions that would benefit students.

Cocaine use, like other illicit drugs, is most often a hidden behavior. The conventional method to measure the extent of drug use in a community is through self-reported surveys or data on drug seizures. Drug seizures are not sufficiently frequent to provide useful information on the prevalence of drug use in college and university communities, and many issues exist with self-reported survey methods [14]. For example, surveys are subject to numerous forms of bias, specifically social

desirability, recall, and nonresponse bias. College and university students may be apprehensive to disclose drug use even when confidentiality is assured due to fear of consequences (academic or legal discipline), privacy concerns, associated stigma, or self-denial [15]. Additionally, surveys are expensive. As a complement to survey and drug seizure data, public health departments outside the U.S. have used wastewater-based epidemiology (WBE) to inform their understanding of trends in drug use since 2011 [16].

When applied to drug use, WBE involves measuring the amount of both drug residues and metabolites (collectively known as biomarkers) in a community's wastewater. Upon ingestion, a drug may exit the body in the form of the parent substance or a metabolite through either urine or feces and subsequently enter the sewage system [17]. Since shown to be possible in the early 2000s [18], WBE has been suggested a powerful tool for monitoring illicit drug usage patterns [19]. Since then, WBE has been successfully deployed to track illicit drug use in Australia [20], Europe [21], South Africa [22], and South Korea [23]. Yet, this approach to monitoring drug use is still considered to be relatively new and has not been widely adopted outside of these contexts. Interestingly, WBE was widely used to support university and college responses to the COVID-19 pandemic [24]; however, WBE of drug use has seen a limited application at institutions of higher learning in the U.S. A few examples include the quantification of non-medical use of ADHD medication [25]; the use of psycho-active stimulants during periods of high and low stress [26]; and documenting the presence of amphetamines, opioids, cocaine, cannabinoids, fentanyl, and lysergics in college wastewater [27–29].

As universities and colleges have a vested interest in the health and wellbeing of their students, including drug use behavior, it would be beneficial to better understand how WBE might compare and complement standard questionnaires about drug use in a university setting. Moreover, WBE not only has relevance for post-secondary intuitions, but also a broader social and public health significance. Herein, we compare a survey of self-reported frequency of cocaine use with the levels of cocaine metabolites (benzoylecgonine) found in wastewater at a university in the U.S. during spring semester 2021.

## Methods

### Setting

This study was conducted at a large university that typically enrolls about 20,000 students, the vast majority (70%) of which are undergraduate students, with over 65% of the student population living in university housing on the main campus. The residence halls are comprised of traditional style college dormitories.

### Cocaine use survey

As part of a larger survey of student well-being implemented by the university between 05/04/2021 and 25/04/2021, we assessed the frequency of cocaine use among students living on campus. The university selected a random sample of the student population and sent a confidential email to the sample of 6,000 undergraduate and graduate students inviting them to participate in the health behavior survey (with an embedded survey link) during spring semester of 2021. The survey received a 33.6% response rate (n = 2013). Students taking the survey were notified that by choosing to complete the survey they were consenting to participate. The survey questionnaire was modeled after the National American College Health Assessment Survey [30]. The questions asked which substances students have used cocaine, their frequency of use, and their general social well-being. No identifying information was collected as part of the survey.

### Wastewater sampling and analysis

During spring semester of 2021 wastewater surveillance was being implemented outside student residence halls (dormitories) as part of the university's COVID-19 response plan [31]. Coinciding with the survey being distributed we pulled fourteen 24-hour composite effluent wastewater samples from 4 dormitory locations across both north and south campuses

for drug analysis, with the number of samples varying by location. (Location 1 was sampled five times, location 2 was sampled six times, location 3 was sampled two times, and location 4 was sampled one time.).

Samples were filtered using 0.22 μm polyethersulfone syringe filters and analyzed in duplicate by liquid chromatography-high-resolution mass spectrometry, as detailed elsewhere [32]. Typically, 1 mL of filtered sample was loaded onto a Hypersil GOLD aQ C18 trap column (20 × 2.1 mm i.d., 12 μm) using a Thermo Scientific TriPlus RSH autosampler and liquid handling system for preconcentration and extraction. Target analytes were then eluted from the trap column for chromatographic separation by a Hypersil GOLD aQ C18 analytical column (100 × 2.1 mm, 1.9 μm; preceded with a 10 × 2.1 mm guard cartridge) on a Vanquish Horizon ultrahigh-performance liquid chromatograph. Mass spectrometric analysis was conducted in positive electrospray ionization mode on an Orbitrap Exploris 240 quadrupole-Orbitrap mass spectrometer. Target analytes were confirmed by comparing their chromatographic retention times and data-dependent mass spectra with those of reference standards. Concentrations of confirmed target analytes were quantified with reference to calibration standards. The limits of quantification for the chemical of interest are as follows: bezoylegconine (1.2 ng/L), acetaminophen (1.3 ng/L), and caffeine (6.3 ng/L) [32].

### Estimating cocaine use from wastewater

We estimated the daily consumption of cocaine, as well as acetaminophen and caffeine, in milligrams per 1,000 population following the approach outlined by Wang et al (Equation 1) [33]. Due to the samples being taken from building estimates, we were unable to measure flow directly. Therefore we estimated flow from the average American wastewater discharge ranging from 190 - 265 liters per day [34]. We obtained correction factors for back calculation from a literature review of current and past WBE studies: acetaminophen 3.55 [35], benzoylecgonine (cocaine metabolite) 3.59 [33], and caffeine 14.8 [36]. The university where the samples were collected provided population counts from each dormitory. Once daily consumption was estimated, we divided the daily consumption estimate by an estimated typical dose of each drug. We used 30–70 mg for cocaine, 500–1000 mg for acetaminophen, and 95–175 mg for caffeine.

$$DC = \left( \frac{C_{DTR} \times F \times CF_{DTR}}{P} \right) / TD$$

(1)

Equation 1: DC is the daily consumption value (mg/1000 people). $C_{DTR}$ is the median concentration of the drug target residue or DTR (ng L-1) from the sample analysis. F is the flow rate (in liters per/day/per person). $CF_{DTR}$ is the correction factor [33] of the indicated DTR. P is the population size indicated to us by the University housing office. TD is the typical dose of each substance represented in mg per day or assuming one dose (benzoylecgonine).

### Calculations of survey data

The survey first asked students the following questions about their use of cocaine a.) to determine if potential lifetime use: "Have you ever used - Cocaine (coke, crack, etc.); b.) to estimate recent use: "Since the start of the spring semester, how often have you used: - Cocaine (coke, crack, etc.). If students answered yes to the second question, they were asked c.) "Since the start of the spring semester, how often have you had a strong desire or urge to use the following" and responded using this Likert scale of frequency, "less than weekly", "weekly", "multiple times per week", and "daily." We classified students answering "multiple times per week" or "daily" as students using cocaine at the time of the wastewater sampling.

### Ethics

This study was reviewed and found exempt by the Syracuse University IRB #21-003. A random sample of 6,000 students were invited by email to participate in a survey on student substance use, health, and well-being. The email included

written informed consent which explained that no names would be obtained in the consent process to ensure anonymity. The informed consent stated "Involvement in the study is voluntary and anonymous. This means you can choose whether to participate and that you may withdraw from the study at any time without penalty. We are not asking for any personally identifiable information. By continuing with this survey, you confirm that you are 18 years of age or older and that you agree to participate in this research study." The invited participants had the option to proceed to the survey or exit. No identifying information was gathered during the survey. The survey was conducted between 05/04/2021 and 21/04/2021.

## Results

The survey was sent to 6,000 students including undergraduates and graduates, with a response rate of 33.6% (n = 2,013). Among the respondents 38% (n = 762) reported living in a campus residential dormitory. We used this segment of the survey respondents to estimate the self-report prevalence of recent cocaine use because the wastewater samples were collected from residence hall outflows. Among students living on campus cocaine use was uncommon, with more than 95% of students reporting they had never used cocaine before. Among students living in campus housing (Fig 1), 3% reported cocaine use during the semester the study was conducted. Only one student was classified as a regular user, reporting cocaine use multiple times per week, and one other student reported using cocaine weekly.

The cocaine metabolite benzoylecgonine was found in all wastewater samples across the four dormitory sampling locations (Fig 2). Raw concentrations were low, less than 100 ng/L. Acetaminophen and caffeine were also found in all wastewater samples across the four dormitory sampling locations, with concentrations 800 and 100 times greater than benzoylecgonine, respectively. Acetaminophen had the highest variability of recorded results, which translated into wide

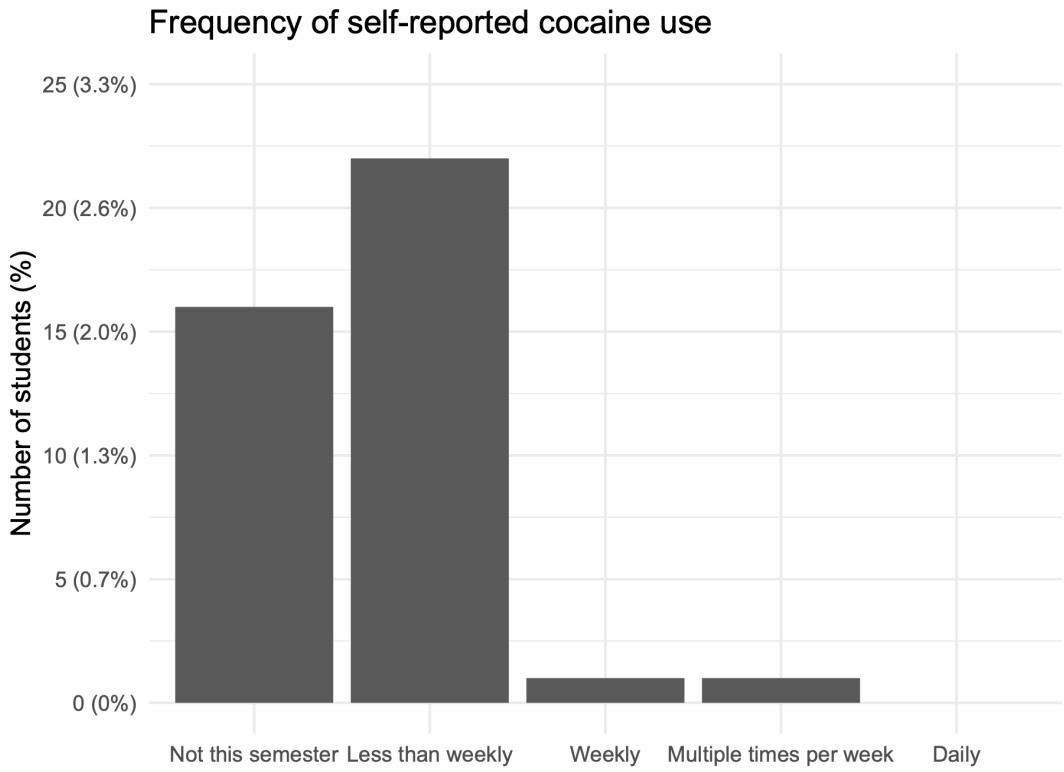

**Fig 1. Frequency of self-reported cocaine use among students living on campus who reported they had ever used cocaine.** 95% of students reported never using cocaine.

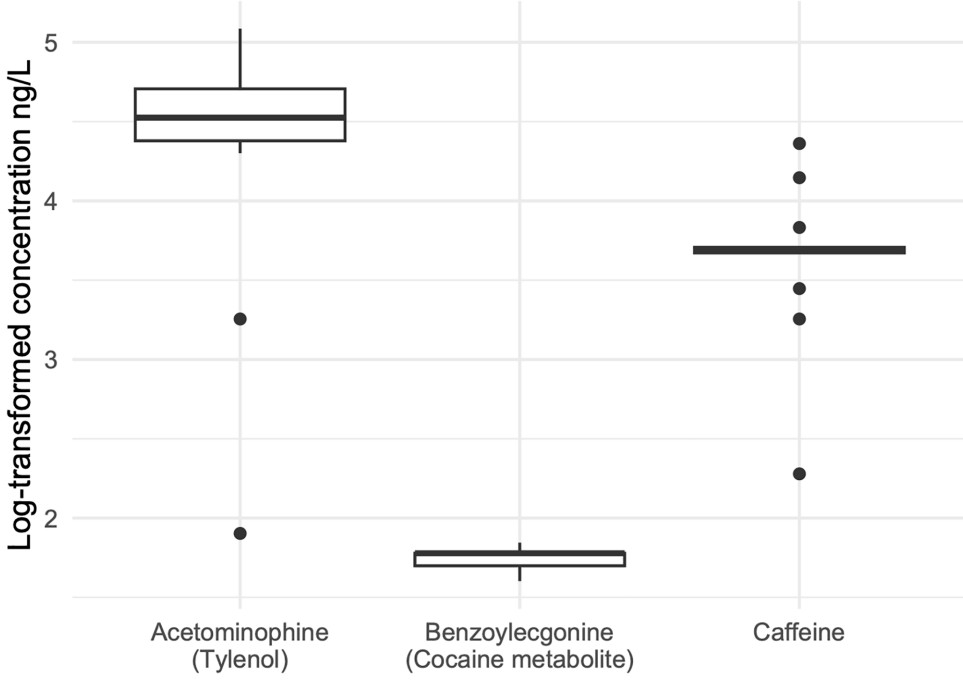

**Fig 2. Raw concentrations of the cocaine metabolite benzoylecgonine, acetaminophen, and caffeine in wastewater from four sampling points across the university campus.**

intervals of estimated users (Table 1). The reported prevalence of caffeine use was less variable but still wide, ranging from 7% − 37%. Cocaine use was estimated to be < 0.5% prevalent at all sampling locations. A total of 4.67 (95% CI = 2.18 = 7.14) doses of cocaine use was measured across all sampling locations, translating to 0.12% of students on campus using cocaine (95% CI = 0.05–0.18%).

## Discussion

In our analysis, we found that self-reported survey estimates of cocaine use among those who reported frequent regular use presented similarly to point estimates of cocaine use derived from wastewater-based epidemiology. The self-report survey results indicated according to our criteria only 1 out of 762 students to be a regular cocaine user (0.13%). This result was equivalent to the estimate of 0.12% of students regularly using cocaine as measured through wastewater-based epidemiology. A prevalence of use of 0.12-0.13% is in line with a recent American College Health Association (ACHA), National College Assessment III [37], which found regular cocaine use across college campuses to be 0.14%.

The use of acetaminophen and caffeine derived from wastewater-based epidemiology were much more variable than we expected. A recent study of the use of pain medications at the city-level showed similar percentages of acetaminophen use to two of the locations we sampled [35], however the other two locations we sampled had estimated prevalence of acetaminophen use well below 1%. These two sampling locations also had the lowest estimated caffeine and cocaine use, perhaps indicating an issue in the sampling locations themselves.

Given the sample size of the ACHA data set, seeing that our results were close to what they found cocaine use to be, is very encouraging but could have also happened by chance. In our data analysis, we only had one person during spring semester classified as a cocaine "regular user." With only 1 user out of 762, we were not able to perform any kind of advanced statistical analyses to identify risk factors with current cocaine use. Another limitation of our study were the

**Table 1. The number of estimated doses and population prevalence of the use of acetaminophen, benzoylecgonine, and caffeine derived through wastewater-based epidemiology.**

| Drug | Location | Number of samples | Median raw concentration (ng/L); (min-max) | Number of estimated doses (credibility interval) | Population prevalence assuming 1 dose per person (credibility interval) |
|---|---|---|---|---|---|
| Acetaminophen | Loc 1 | 5 | 27,000 (80–94,000) | 35 (18–51) | 9.5% (5.0–14%) |
| Acetaminophen | Loc 2 | 6 | 49,000 (33,000–122,000) | 63 (33–92) | 12.8% (6.8–18.9%) |
| Acetaminophen | Loc 3 | 2 | 26,500 (20,000–33,000) | 34 (18–50) | 1.7% (0.9–2.5%) |
| Acetaminophen | Loc 4 | 1 | 1,800 | 2.3 (1.2–3.4) | 0.2% (0.1–0.3%) |
| Benzoylecgonine (cocaine metabolite) | Loc 1 | 5 | 50 (40–70) | 1.04 (0.49–1.59) | 0.3% (0.1–0.4%) |
| Benzoylecgonine (cocaine metabolite) | Loc 2 | 6 | 55 (50–60) | 1.14 (0.53–1.75) | 0.2% (0.1–0.4%) |
| Benzoylecgonine (cocaine metabolite) | Loc 3 | 2 | 60 (60–60) | 1.24 (0.58–1.90) | 0.06% (0.03–0.09%) |
| Benzoylecgonine (cocaine metabolite) | Loc 4 | 1 | 60 | 1.24 (0.58–1.90) | 0.1% (0.05–0.17%) |
| Caffeine | Loc 1 | 5 | 4700 (190–5,100) | 135 (75–194) | 37% (0.21–0.54%) |
| Caffeine | Loc 2 | 6 | 4700 (2,800–23,000) | 135 (75–194) | 28% (15–40%) |
| Caffeine | Loc 3 | 2 | 4900 (4,700–5,100) | 141 (79–203) | 7.0% (3.9–10.1%) |
| Caffeine | Loc 4 | 1 | 6800 | 195 (109–281) | 18% (9.9–26%) |

various assumptions that had to be made in our initial daily consumption (DC) equation. These assumptions included the median flow rate (F) (which was based off a government study from 2002) and the typical dose (TD) for each substance (i.e., the dose size or frequency of cocaine administration). We inferred a typical dose size but not have any data from our study. Additionally, the survey asked about cocaine use throughout the semester, and we only tested wastewater for cocaine use during the survey period. This temporal misalignment precluded us from testing wastewater for known cocaine use at the time of the survey. The agreements between WBE and self-reported cocaine use that we've observed could be coincidental.

As in any WBE study, uncertainties revolve around establishing a standardized correction factor for DTRs, considering the loss of biomarkers in metabolic processing as well as additional interactions with different substances in sewer systems [38]. Our study design was cross sectional and only captured usage data from a single period to determine the exposure of drug use on campus. We are limited by the number of samples from spring semester 2021 and we cannot infer spatial or temporal patterns from this data, nor did we attempt any investigation into the reason why cocaine is being used on this campus [39]. Considering the study design and implementation, our study is perhaps not representative of the entire campus.

The strengths of wastewater-based epidemiology allow researchers to obtain drug use information without breaching the personal privacy of students and avoid the issues which can be encountered with self-report survey-based research. In a recent meta-analysis examining the relationship between self-reported drug use and wastewater samples [40], it was found that the agreement between the two measures were consistently high, suggesting both are equally good methods of obtaining reliable data. The relationship tended to be stronger when research participants were informed they would not face any repercussions for reporting drug use, with an emphasis on cocaine usage.

This study confirmed the viability of WBE as a public health tool to complement existing methods of obtaining drug use information, particularly on college campuses. Future research could expand to multiple drugs of interest such as various opioids, cannabis, and amphetamine type stimulants. It would also be pertinent to have a flow volume which is specific to the population rather than a best approximation based on previous studies. If this study were to be replicated, multiple wastewater sample collections across the semester or academic year and would enable researchers to gain a better

understanding of student population level cocaine use. Collecting samples over time (ex. monthly over the semester) may better capture the population level of cocaine use because cocaine's metabolite benzoylecgonine is only detectable in urine for about 72–98 hours after last use [41]. Collecting and analyzing wastewater samples over the course of a semester may yield an estimate closer to the residential student population self-reported usage rate of 3% during the semester.

WBE can be a useful to obtain noninvasive drug use data, and with this in mind, future research should adopt a longitudinal study design to evaluate spatial and temporal patterns of use. Having a tool which can give colleges and universities an idea of temporal patterns of use, may enable them to design appropriate strengths-based interventions, to optimize student mental health and wellbeing. It is important then when taking samples from dormitories and residence halls on campus, that the data be handled with caution, as the mishandling of such data may inadvertently stigmatize a certain residence where high drug usage may be detected.

## Supporting information

**S1 File. Wastewater data.**
(XLSX)

**S2 File. Survey data.**
(XLSX)

## Acknowledgments

We thank Dr. Shiru Wang (Syracuse University) for her contribution to sample analysis. This study was approved by Syracuse University's IRB. IRB # 21-003.

## Author contributions

**Conceptualization:** Dessa Bergen-Cico, David A. Larsen.

**Data curation:** Shona McCulloch.

**Formal analysis:** Shona McCulloch, Dessa Bergen-Cico, Teng Zeng, David A. Larsen.

**Investigation:** Shona McCulloch, Dessa Bergen-Cico, Teng Zeng.

**Methodology:** Shona McCulloch, Teng Zeng.

**Project administration:** Dessa Bergen-Cico, David A. Larsen.

**Resources:** Dessa Bergen-Cico, Teng Zeng.

**Supervision:** David A. Larsen.

**Validation:** Dessa Bergen-Cico.

**Visualization:** Shona McCulloch, David A. Larsen.

**Writing – original draft:** Shona McCulloch, David A. Larsen.

**Writing – review & editing:** Dessa Bergen-Cico, Teng Zeng, David A. Larsen.

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
