## [Decision Letter · Decision Letter 0]

8 Oct 2025

Dear Dr. Larsen,

Thank you for submitting your manuscript to PLOS ONE. After careful consideration, we feel that it has merit but does not fully meet PLOS ONE’s publication criteria as it currently stands. Therefore, we invite you to submit a revised version of the manuscript that addresses the points raised during the review process.

We look forward to receiving your revised manuscript.

Kind regards,

Enrico Greco

Academic Editor

PLOS ONE

Journal Requirements:

When submitting your revision, we need you to address these additional requirements

 “TZ received funding from the National Science Foundation, award number 2018497” 

4. Please note that funding information should not appear in any section or other areas of your manuscript. We will only publish funding information present in the Funding Statement section of the online submission form. Please remove any funding-related text from the manuscript.

Reviewer's Responses to Questions

**Comments to the Author**

1. Is the manuscript technically sound, and do the data support the conclusions?

Reviewer #1: Partly

Reviewer #2: Yes

2. Has the statistical analysis been performed appropriately and rigorously?

Reviewer #1: No

Reviewer #2: Yes

3. Have the authors made all data underlying the findings in their manuscript fully available?

Reviewer #1: Yes

Reviewer #2: No

4. Is the manuscript presented in an intelligible fashion and written in standard English?

Reviewer #1: Yes

Reviewer #2: Yes

Reviewer #1: The authors compare estimate of cocaine use from benzoylecgonine concentrations in 14 24-hour composite wastewater samples collected from four dormitory sewages with results from a student self-report survey (n = 2,013 respondents; 762 dormitory residents). They conclude that wastewater-based epidemiology (WBE) and self-reported data give similar prevalence estimates (~ 0.12 - 0.13%). The manuscript is technically competent in the laboratory analysis, but the statistical treatment and handling of uncertainty in the comparison require strengthening.

Specific comments:

1. Abstract:

in the conclusion, the sentence “WBE is a feasible and valid of method” should be replaced with a more cautious statement, such as: “WBE shows promise as a complementary approach; with current data the WBE point estimate is similar to the survey point estimate, but uncertainty in both measures (especially the survey) is large, and further work is required.”

2. The extended possessive form is preferable:

Line 28: “…on the brain’s dopaminergic pathways…”

Line 36: “…world’s largest…”

Line 66: “…community’s wastewater…”

Line 104: “…university’s COVID-19…”

Line 221: “…cocaine’s metabolite…”

Line 224: “…residential population’s self-reported usage…”

3. Line 39: There is an error in the reference “[UNODC, 2023]”

4. Line 39: missing period in “U.S.”

5. Line 42-44: university life is not the only stressor, according to reference [9]

6. Line 62-64: the reference for this sentence is missing

7. Line 79-84: The aim of the study could be expanded. Specifically, the manuscript could test whether WBE is truly capable of providing an overview of cocaine use within college settings. Moreover, this issue has not only institutional relevance for universities and colleges, but also broader social and public health significance.

8. “Cocaine use survey” section:

The manuscript should clarify how many students received the survey invitation email. This information should be included not only in the “Results” section but also in the “Materials and methods”.

9. “Wastewater sampling and analysis” section:

The manuscript should specify how the 14 are distributed across the four dormitories. In addition, it would be valuable to report the number of residents in each dormitory, in order to assess whether these are comparable.

Comments:

10. Comparative framework:

The comparison of single-point WBE estimates (14 daily composites across four locations) with survey responses covering an entire semester is problematic. The survey asks about behavior “since the start of the spring semester” and defines regular users as those reporting use “multiple times per week/daily.” By contrast, WBE captures only a limited number of 24-hour composites. This temporal mismatch, combined with sparse WBE sampling, introduces considerable variability: benzoylecgonine is detectable for approximately 3–4 days post-use, and a small number of users can substantially affect daily loads. The authors should acknowledge that observed agreement between methods could be coincidental, given the short WBE sampling window.

The paper states that estimates “aligned”, but given the methodological imprecision described above, the authors should either (a) present explicit overlapping confidence intervals, or (b) conduct a probabilistic equivalence test. As presented, the textual claim of equivalence is weak.

11. Avoid over-strong language by replacing words such as “aligned” / “equivalent” with a statement like: “Point estimates are similar, but the survey estimate is imprecise (wide CI) and agreement is sensitive to key WBE assumptions; thus results are consistent but not conclusive.” Include both intervals in the Discussion section.

Overall evaluation:

The study is conceptually strong, methodologically promising, and well-motivated. However, the current statistical treatment of uncertainty and the strength of the comparison between WBE and survey data are limited. Incorporating the suggested clarifications, explicitly presenting uncertainty intervals, and adopting a more cautious interpretation would substantially strengthen the manuscript and allow the claims to be presented more transparently.

Reviewer #2: The proposed paper of McCulloch and co-workers presents an interesting case study of wastewater-based epidemiology; significant weakpoints are present, both in the model for estimating drug dose consumption - presenting a number of assumptions - and in extension of sampling; authors are well aware of such points and highlight them as points of attention. I ask (1) to report in the text Limit of Quantification for benzoylegconine, acetaminophen and caffeine).- and (2) at least in the supplemental material - a table with complete basic statistics of 14 samples for each sampling site for the three considered analytes (n, average, min, max).

**Do you want your identity to be public for this peer review?** For information about this choice, including consent withdrawal, please see our Privacy Policy

Reviewer #1: No

Reviewer #2: No

---

## [Author Response · Author response to Decision Letter 1]

25 Nov 2025

Thank you for the review of our manuscript. We have responded to each of the reviewers' questions and suggestions. Their points are copied here with our response directly underneath.

Editor:

Thank you for the guidance. We have updated the manuscript according to the style guides.

We have updated the manuscript with the ethics information required. We have included the following paragraph:

“A random sample of 6,000 students were invited by email to participate in a survey on student substance use, health, and well-being. The email included written informed consent which explained that no names would be obtained in the consent process to ensure anonymity. The informed consent stated: “Involvement in the study is voluntary and anonymous. This means you can choose whether to participate and that you may withdraw from the study at any time without penalty. We are not asking for any personally identifiable information. By continuing with this survey, you confirm that you are 18 years of age or older and that you agree to participate in this research study.”

“TZ received funding from the National Science Foundation, award number 2018497”

We have updated the financial disclosure as indicated.

4. Please note that funding information should not appear in any section or other areas of your manuscript. We will only publish funding information present in the Funding Statement section of the online submission form. Please remove any funding-related text from the manuscript.

Thank you for the guidance. We have removed the funding statement from the manuscript.

Thank you for the guidance. All authors are agreed in making the data publicly available.

Thank you for the guidance. We do not have any supporting information files.

Thank you for the guidance. The reviewers did not recommend any citations.

Reviewer #1: The authors compare estimate of cocaine use from benzoylecgonine concentrations in 14 24-hour composite wastewater samples collected from four dormitory sewages with results from a student self-report survey (n = 2,013 respondents; 762 dormitory residents). They conclude that wastewater-based epidemiology (WBE) and self-reported data give similar prevalence estimates (~ 0.12 - 0.13%). The manuscript is technically competent in the laboratory analysis, but the statistical treatment and handling of uncertainty in the comparison require strengthening. Specific comments:

1. Abstract:

In the conclusion, the sentence “WBE is a feasible and valid of method” should be replaced with a more cautious statement, such as: “WBE shows promise as a complementary approach; with current data the WBE point estimate is similar to the survey point estimate, but uncertainty in both measures (especially the survey) is large, and further work is required.”

Thank you for the recommendation. We have replaced the sentence “WBE is a feasible and valid of method,” with the recommended cautious statement: “WBE shows promise as a complementary approach; with current data the WBE point estimate is similar to the survey point estimate, but uncertainty in both measures (especially the survey) is large, and further work is required.”

2. The extended possessive form is preferable:

Line 28: “…on the brain’s dopaminergic pathways…”

Cocaine is an illicit stimulant drug with a high addiction potential due to its effects on the pathways of the [DAL1] brain’s dopaminergic system [1].”

Line 36: “…world’s largest…”

“Historically, North America has been the largest market in the world for cocaine with an estimated 6.4 million users in 2020;”

Line 66: “…community’s wastewater…”

Line 104: “…university’s COVID-19…”

Line 221: “…cocaine’s metabolite…”

Line 224: “…residential population’s self-reported usage…”

Thank you for the recommendation. We have a difference in writing style from the reviewer. We have gone ahead and changed some of these but not others.

3. Line 39: There is an error in the reference “[UNODC, 2023]”

Thank you for catching this error. We’ve removed the “[UNODC, 2023]” – that reference is captured as reference 5 in the manuscript.

4. Line 39: missing period in “U.S.”

Thank you for catching this. We have added the missing period to the “U.S.”

5. Line 42-44: university life is not the only stressor, according to reference [9]

Yes, we agree with the reviewer that there multiple stressors. We have revised this sentence to state that university life presents multiple stressors.

6. Line 62-64: the reference for this sentence is missing

Thank you for the recommendation. We have revised this sentence to show that this is primarily ocurring outside the US and cited a longitudinal study from the SCORE network.

7. Line 79-84: The aim of the study could be expanded. Specifically, the manuscript could test whether WBE is truly capable of providing an overview of cocaine use within college settings. Moreover, this issue has not only institutional relevance for universities and colleges, but also broader social and public health significance.

Thank you for the suggestion. Unfortunately we determined the aim of the study before data collection and writing the manuscript. Herein we can only how comparable WBE is to self reported survey measures. We have added a new sentence to the paragraph in question: “Moreover, WBE not only has relevance for post-secondary institutions, but also a broader social and public health significance.”

8. “Cocaine use survey” section:

The manuscript should clarify how many students received the survey invitation email. This information should be included not only in the “Results” section but also in the “Materials and methods”.

Thank you for the recommendation. We have added the number of students receiving the survey as suggested.

9. “Wastewater sampling and analysis” section:

The manuscript should specify how the 4 are distributed across the four dormitories. In addition, it would be valuable to report the number of residents in each dormitory, in order to assess whether these are comparable.

Thank you for the suggestions. We’ve added the detail on how many samples per location. Unfortunately the number of residents in each dormitory would identify the dormitory at Syracuse University and so we cannot report on that information.

10. Comparative framework:

The comparison of single-point WBE estimates (14 daily composites across four locations) with survey responses covering an entire semester is problematic. The survey asks about behavior “since the start of the spring semester” and defines regular users as those reporting use “multiple times per week/daily.” By contrast, WBE captures only a limited number of 24-hour composites. This temporal mismatch, combined with sparse WBE sampling, introduces considerable variability: benzoylecgonine is detectable for approximately 3–4 days post-use, and a small number of users can substantially affect daily loads. The authors should acknowledge that observed agreement between methods could be coincidental, given the short WBE sampling window.

The paper states that estimates “aligned”, but given the methodological imprecision described above, the authors should either (a) present explicit overlapping confidence intervals, or (b) conduct a probabilistic equivalence test. As presented, the textual claim of equivalence is weak.

Thank you for the recommendation. This is certainly a limitation to the study and we have added this limitation to our limitations paragraph. We have also revised the first sentence of the discussion to say “presented similarly” rather than “alligned”. The limitations paragraph now has the following:

“Additionally, the survey asked about cocaine use throughout the semester, and we only tested wastewater for cocaine use during the survey period. This temporal misalignment precluded us from testing wastewater for known cocaine use at the time of the survey. The agreements between WBE and self-reported cocaine use that we’ve observed could be coincidental.”

11. Avoid over-strong language by replacing words such as “aligned” / “equivalent” with a statement like: “Point estimates are similar, but the survey estimate is imprecise (wide CI) and agreement is sensitive to key WBE assumptions; thus results are consistent but not conclusive.” Include both intervals in the Discussion section.

Thank you for the recommendations. We’ve replaced the term “aligned” with the following sentence in the “Results” summary in the abstract: “Self-reported survey estimates of cocaine use and point estimates of cocaine use derived from wastewater-based epidemiology are similar, but the survey is imprecise with a wide CI, and agreement is sensitive to key WBE assumptions; thus, results are consistent but not conclusive.” We also replaced the term “aligned” with “presented similarly to,” in the “Discussion” section.

Overall evaluation: The study is conceptually strong, methodologically promising, and well-motivated. However, the current statistical treatment of uncertainty and the strength of the comparison between WBE and survey data are limited. Incorporating the suggested clarifications, explicitly presenting uncertainty intervals, and adopting a more cautious interpretation would substantially strengthen the manuscript and allow the claims to be presented more transparently.

Thank you for your time in reviewing our manuscript. We would be happy to further revise if needed.

Reviewer #2: The proposed paper of McCulloch and co-workers presents an interesting case study of wastewater-based epidemiology; significant weakpoints are present, both in the model for estimating drug dose consumption - presenting a number of assumptions - and in extension of sampling; authors are well aware of such points and highlight them as points of attention. I ask (1) to report in the text Limit of Quantification for benzoylegconine, acetaminophen and caffeine).- and (2) at least in the supplemental material - a table with complete basic statistics of 14 samples for each sampling site for the three considered analytes (n, average, min, max).

Thank you for the review of our manuscript.

1) We have added a sentence to the methods section with the limits of quantification for each of the chemicals tested.

2) We have added the complete basic statistics for the samples to table 1 as suggested.

---

## [Editor Report · Decision Letter 1]

28 Nov 2025

Comparison of self-reported survey and wastewater-based epidemiology measures of cocaine use on a college campus

PONE-D-25-45699R1

Dear Dr. Larsen,

We’re pleased to inform you that your manuscript has been judged scientifically suitable for publication and will be formally accepted for publication once it meets all outstanding technical requirements.

Kind regards,

Enrico Greco

Academic Editor

PLOS ONE
---

## [Editor Report · Acceptance letter]

PONE-D-25-45699R1

PLOS One

Dear Dr. Larsen,

I'm pleased to inform you that your manuscript has been deemed suitable for publication in PLOS One. Congratulations! Your manuscript is now being handed over to our production team.

Kind regards,

on behalf of

Dr. Enrico Greco

Academic Editor

PLOS One